# SHOW YOUR WORK: SCRATCHPADS FOR INTERMEDIATE COMPUTATION WITH LANGUAGE MODELS

## ABSTRACT

Large pre-trained language models perform remarkably well on tasks that can be done "in one pass", such as generating realistic text (Brown et al., 2020) or synthesizing computer programs (Chen et al., 2021; Austin et al., 2021). However, they struggle with tasks that require unbounded multi-step computation, such as adding integers (Brown et al., 2020) or *executing* programs (Austin et al., 2021). Surprisingly, we find that these same models are able to perform complex multi-step computations—even in the few-shot regime—when asked to perform the operation "step by step", showing the results of intermediate computations. In particular, we train Transformers to perform multi-step computations by asking them to emit intermediate computation steps into a "scratchpad". On a series of increasingly complex tasks ranging from long addition to the execution of arbitrary programs, we show that scratchpads dramatically improve the ability of language models to perform multi-step computations.

## 1 INTRODUCTION

Large Transformer-based language models exhibit surprisingly impressive capabilities (Devlin et al., 2019; Brown et al., 2020), including the ability to generate code that solves simple programming problems (Chen et al., 2021; Austin et al., 2021). However, these models struggle to perform multi-step algorithmic calculations, especially those that require precise reasoning and unbounded computation. For example, GPT-3 struggles to perform few-shot addition on numbers with greater than three digits (Brown et al., 2020). Similarly, large-scale language models struggle to predict the result of executing Python code, even code which is a solution to a programming task the model is able to solve (Austin et al., 2021). Likewise, standard recurrent and graph neural networks fail to systematically generalize when predicting the output of simple programs with loops (Bieber et al., 2020). So language models can to some extent *write* code, but do not seem to accurately represent the semantics of the code they write, because they cannot predict its execution. This has motivated research on networks that can perform algorithmic reasoning (Graves et al., 2014; Zaremba & Sutskever, 2014; Bieber et al., 2020). Neural networks that accurately represent the semantics of programs could enable a variety of downstream tasks, including program synthesis (Devlin et al., 2017), program analysis (Allamanis et al., 2018), and other algorithmic reasoning tasks (Velickovic & Blundell, 2021).

Why do large language models struggle with algorithmic reasoning tasks? We suggest that this is at least partly due to a limitation of the way the Transformer architecture is applied to these tasks: the model is asked to perform these tasks in one forward pass. Given a fixed number of layers and a fixed amount of computation time, the model cannot adapt the amount of compute spent on a problem to its difficulty before producing an output.[1] Prior work (Graves, 2016; Banino et al., 2021) has explored neural architectures that explicitly allow for dynamically chosen amounts of computation time to be dedicated to different sub-tasks. In this work, we propose a different approach—one that

---

[1]Transformers perform a computation which is quadratic in the length of the input sequence, so are theoretically unable to perfectly simulate algorithms which have greater time complexity than $O(n^2)$. However, it is unclear how relevant this theoretical bound is in practice; neural sequence prediction is approximate, and Transformers may be large enough in practice to effectively memorize the correct solutions for a relevant subspace of the possible inputs (e.g., all inputs up to a certain size).

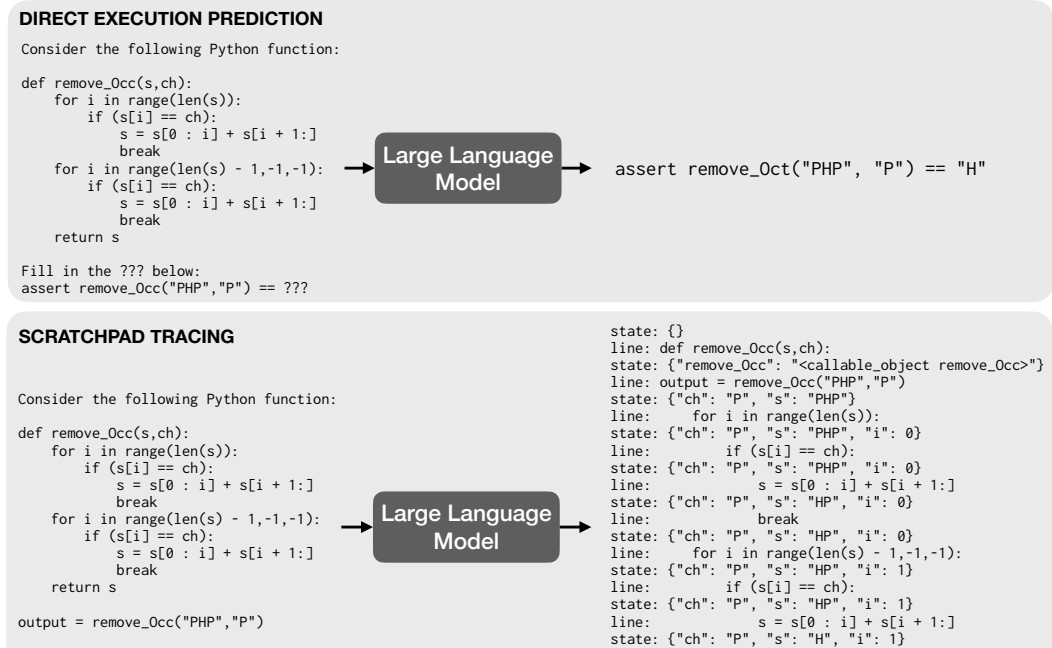

Figure 1: Overview of our scratchpad approach applied to predicting code execution and comparison to direct execution prediction. Top: Previous work has shown that large pre-trained models achieve poor performance when asked to directly predict the result of executing given computer code (Austin et al., 2021). Bottom: In this work, we show that training models to use a *scratchpad* and predict the program execution trace line-by-line can lead to large improvements in execution prediction performance. N.B. Although the example above only has one loop iteration for each loop, all loops are unrolled across time.

can exploit existing Transformer architectures and large few-shot-capable language models—we modify the *task design* rather than the model or training procedure.

Our proposal is simple: Allow the model to produce an arbitrary sequence of intermediate tokens, which we call a *scratchpad*, before producing the final answer. For example, on addition problems, the scratchpad contains the intermediate results from a standard long addition algorithm (see Figure 2). To train the model, we encode the intermediate steps of the algorithm as text and use standard supervised training.

This paper makes the following contributions:

- We introduce (Section 2) the notion of a "scratchpad" for Transformers, in order to make them better at performing complex discrete computations without modifying the underlying architecture.

- We show (Section 3) that scratchpads help Transformers learn to perform long addition in the fine-tuning regime, and in particular that they improve out-of-distribution generalization to larger problem instances.

- We also find (Section 4) that scratchpads help Transformers perform a somewhat higher level task: polynomial evaluation. This is true in both the few-shot and fine-tuning regimes.

- Finally, we move to a much more general context and show (Section 5) that training Transformers to emit full program traces line by line annotated with local variables dramatically improves their ability to predict the result of executing a given computer program on a particular input. This application in some sense subsumes the others.

## 2 METHOD

In this work we consider two related problems: algorithm induction (Graves et al., 2014; 2016; Kurach et al., 2016; Kaiser & Sutskever, 2016) and learning to execute (Zaremba & Sutskever, 2014; Bieber et al., 2020). The goal of both problems is for the neural network to learn to emulate a function $f$, which is "algorithmic" in the sense that it can be represented by a short program, such as addition or polynomial evaluation, from input-output behavior. In *neural algorithm induction*, the goal is to learn a single algorithm, and each training example gives a single input and desired output represented as strings. Therefore, the training data is $D = \{x_i, f(x_i)\}_{i=1}^N$. For *learning to execute*, we want the model to produce the result of a program, represented as source code, on some input. If each $\pi_i$ is the source code of a program $f_i$, then the training data is $D = \{(\pi_i, x_i, f_i(x_i))\}_{i=1}^N$ (it is common for each $f_i$ to have multiple input-output examples, but we omit this to lighten notation).

The main idea of this paper is that to solve a given algorithmic task, we simply encode the intermediate steps of the algorithm as text and train the model to emit them to a buffer that we call a "scratchpad." For example, let us consider the algorithmic induction task of learning long addition. To teach a model to add 29 to 57, a training example may look like the text in Figure 2, where the steps of the grade-school long addition algorithm are written out explicitly.

Learning to execute tasks can be encoded in a similar way, except now we add the source code $\pi_i$ before the input, scratchpad, and desired output. An example of a training example for a learning to execute task is shown in Figure 1.

At training time, the model will be given the input plus target for standard likelihood-based training. At test time, the model will be given only the input and will be required to predict the target, e.g., by beam search or temperature sampling. In principle, any sequence model could be used for this. In this work, we choose to use decoder-only Transformer language models, but other sequence models could be effective, such as encoder-decoder models (Raffel et al., 2019), or recurrent networks.

```
Input:
2 9 + 5 7

Target:
<scratch>
2 9 + 5 7 ,  C: 0
2 + 5 , 6 C: 1  # added 9 + 7 = 6 carry 1
, 8 6 C: 0  # added 2 + 5 + 1 = 8 carry 0
0 8 6
</scratch>
8 6
```

Figure 2: Example of input and target for addition with a scratchpad. The carry is recorded in the digit following "C:". Comments (marked by #) are added for clarity and are not part of the target.

Adding a scratchpad has several potential advantages: First, the model has adaptive computation time, that is, it can now process the information for as long as needed, depending on the complexity of the task given the input. Second, the model can store the intermediate state of its computation in the scratch buffer and refer back to it by attending to its context. This removes the need to store all intermediate state in activations. Third, by forcing the model to output concrete intermediate states by sampling from the generative model, we aim to reduce the propagation and compounding of small errors, because states are quantized to token embeddings. Compounded errors can show up in methods—like Neural Turing Machines (Graves et al., 2014)—that use recurrence to support extended computations. Finally, examining a model's scratchpad output can help us identify common errors and correct them by revising the scratchpad format. We found this ability to interpret errors to be useful in this work.

For all experiments, we used pre-trained dense decoder-only Transformer language models, ranging in size from 2 million to around 100 billion parameters (with the largest denoted 100B+). These models were pre-trained on web documents and dialog data. We omit some details here—including exact model sizes—for double-blind review, but will include specific details in the published version.

## 3 ADDITION

As a first task, we consider integer addition. The baseline addition task presents two numbers as the input, and the target is their sum. For example:[2]

---

[2]We introduce spaces between the digits to ensure that each digit is mapped to a separate token.

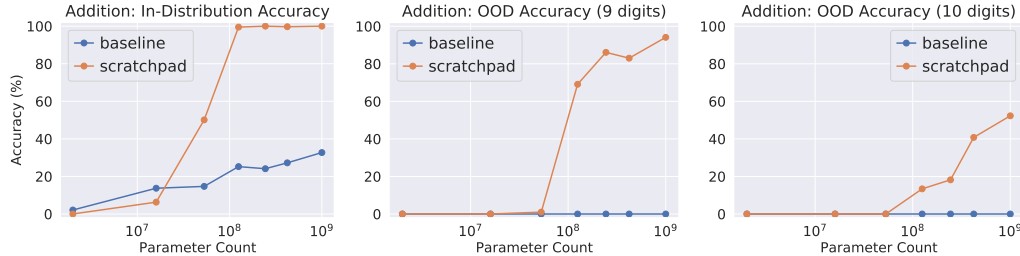

Figure 3: Using a scratchpad significantly improves the performance of pre-trained Transformer-based models on addition, including their ability to generalize out of the training distribution to numbers with more digits. Models were trained on 1-8 digit addition. The baseline models were trained without intermediate scratchpad steps.

```
Input:  2 9 + 5 7
Target: 8 6
```

We implement the scratchpad by including the intermediate steps of the long addition algorithm in the target, as in Figure 2. We train several models on integer addition problems with inputs that have 1-8 digits. We then test performance on in-distribution addition problems (with up to 8 digit inputs), and on out-of-distribution problems with 9 and 10 digit inputs. The models were fine-tuned on 100k examples for 5k steps with batch size 32. There are 10k in-distribution test examples, and 1k test examples for each out-of-distribution task. We examine the performance as a function of model size, ranging from 2M to 1B parameters. We compare performance to a baseline which includes the input and target numbers, but no intermediate scratchpad steps.

**Results**    Figure 3 compares the performance of the scratchpad algorithm with the baseline. We see that beyond a critical model size, models are able to solve the addition task using the scratchpad, while models trained without a scratchpad fail to do so even at the largest tested scale. On the out-of-distribution tasks (9-10 digit addition), we find that models trained without scratchpad completely fail, while models trained with scratchpad show consistent improvement as a function of model size.

## 4    POLYNOMIAL EVALUATION

Next we focus on a slightly higher-level task: evaluating polynomials. Inspired by the "polynomial evaluation" subproblem in Saxton et al. (2019), we generate a dataset of polynomials of degree less than or equal to three, with integer coefficients and inputs constrained to the range $[-10, 10]$. We also restrict outputs to the range $[-1000, 1000]$. We generate a training dataset of 10,000 polynomials and a test dataset of size 2,000. An example scratchpad target for this task is shown in Figure 4, with each term of the polynomial evaluated separately. As in the previous section, we compare the results of direct execution with the results of using the scratchpad. In this experiment, we evaluate in the few-shot regime using a 100B+ parameter pre-trained decoder-only model, as previous work indicates that very large models may be able to perform additions and multiplications with 3 or fewer digits few-shot (Brown et al., 2020). We use $n = 4$ example problems in the few-shot prompt. We also evaluate in the fine-tuning regime with an 8B parameter model fine-tuned for 2000 steps on the training set. The results of both evaluations are shown in Table 1. We find that scratchpad execution outperforms direct execution significantly in both the few-shot and fine-tuning regimes.

## 5    EXECUTING PYTHON PROGRAMS

We have shown that scratchpads can help algorithm induction, that is, they can help models learn to implement a particular algorithm with direct algorithm-specific supervision. But needing to hand-design the intermediate states for every new task is sub-optimal. In this section, we evaluate whether a model can learn to implement a new algorithm by executing arbitrary code. To test this capability, we follow the problem setup from Austin et al. (2021), in which language models are asked to predict the result of executing a given Python program on a particular input. Language models performed poorly at this task, even on programs which are solutions to a programming tasks the model is able to

```
Input:
Evaluate -7*x**2 + 7*x + 5 at x = 1

Target:
<scratch>
-7*x**2: -7
7*x: 7
5: 5
</scratch>
total: 5
```

Figure 4: Example of polynomial evaluation with a scratchpad. Each term in the polynomial is computed separately and then added.

Table 1: Results for polynomial evaluation task. Scratchpad outperforms direct prediction whether using fine-tuning or few-shot.

|  | Few-shot | Fine-tuning |
|---|---|---|
| Direct prediction | 8.8% | 31.8% |
| Scratchpad | **20.1%** | **50.7%** |

solve. Here we show that the scratchpad technique can dramatically improve the ability of language models to execute programs.

**Direct execution prediction**  Our main baseline is the direct execution prediction procedure explored in Austin et al. (2021). Models are shown the source code for a function, and asked to predict the output of running the function on specific inputs. For example, the function in Figure 1 takes as input a string s and a character ch, and removes the first and last instances of the character ch from the string s. The direct execution prompt and target for this task are shown in the "Direct Execution Prediction" box in Figure 1. A task is considered solved under this regime if the model correctly outputs the target string.

**Execution prediction via scratchpad tracing**  As discussed above, direct execution prediction requires the model to correctly output the result of executing the entire function in a single pass. Direct execution prediction has been shown to perform poorly on Python programs in Austin et al. (2021). We therefore design a scratchpad formulation of the execution task, in which models predict the output of a program by first predicting the sequence of intermediate states computed during the program's execution. Formally, we train models to predict an alternating sequence of 1) the ordered sequence of source code lines executed, and 2) the state of the local variables after each line is executed. We call this object the program's **trace**, and it allows us to track both the control flow— the sequence of operations executed—and how the state changes as a result of each operation. We represent the trace as a string, with the line of code reproduced directly, and the state information represented as a JSON dictionary.[3] For example, the "Scratchpad Tracing" box in Figure 1 contains the tracing prompt and trace target for the function discussed above.

Concretely, for each function to be traced, the prompt is formed by printing the function definition, followed by a line which calls the function on a particular input: `output = fn_name(input_value)`, where `fn_name` and `input_value` are replaced with the corresponding function name and input value. In Figure 1, note how the correct output of `remove_Occ("PHP","P")` is shown in the last line of the trace, assigned to the variable `"output"`. A tracing example is considered to have the correct execution output if the encoding of the value assigned to the variable `output` in the last line is a semantic match with the target output value (here, `"output": "P"`). We consider a *task* to be executed correctly if all given input-output examples are correctly executed. We can also test whether there is a "trace exact match" between the model prediction and the ground truth trace, by a) semantically comparing each state in the trace to the corresponding state in the ground truth trace, and b) comparing the sequence of source code lines predicted with the ground truth sequence.

**Experimental setup**  As a proof-of-concept, we first show that scratchpad tracing greatly improves execution performance on synthetic Python data. Then, we compare scratchpad tracing and execution on the human-written Python problems from Austin et al. (2021). We find that a novel data augmentation technique that uses programs generated by the model as additional training data can significantly increase tracing performance on real data, whereas this augmentation technique hurts

---

[3]Some objects cannot be represented using this JSON representation. Some objects (such as tuples) are represented by the closest JSON data type (in this case, lists). Other objects, such as user-constructed objects, are represented by a placeholder string, e.g., `"<object myObject>"`. Functions are also represented as strings, e.g., `"<callable_object f>"`.

```
def f(v0):
    v0 += 0
    v4 = 2
    while v4 > 0:
        v4 -= 1
        v0 *= 2
    return v0

output = f(8)
```

Figure 5: Example synthetic Python program.

Table 2: Synthetic tracing and execution results. Scratchpad outperforms direct prediction both for few-shot and fine-tuned.
*The accuracy criterion for the few-shot scratchpad condition was slightly modified, see the text of Section 5.1 for more details.

|  | Few-shot | Fine-tuned |
|---|---|---|
| Direct prediction | 11% | 20% |
| Scratchpad | 26.5%* | **41.5%** |

performance for direct execution. Finally, we show that incorporating tracing data from additional sources further improves tracing performance, indicating that the scratchpad tracing technique explored here may scale well with more data.

For all experiments on Python code, we use a Transformer model with around 100 billion parameters (100B+), a context window of 1024 tokens and a limit of 512 generation tokens. Unless otherwise stated, all fine-tuning runs used a batch size of 8192 tokens and a learning rate of 3e-5, and model inference was performed with decoding temperature set to $T = 0$, equivalent to greedy decoding.

## 5.1 SCRATCHPAD BEATS DIRECT EXECUTION FOR SYNTHETIC PYTHON PROGRAMS

In our first experiment, we test the few-shot and fine-tuned execution capabilities of our models on simple synthetic Python programs. This provides a proof-of-concept for our tracing technique.

We use a dataset of synthetic Python programs modified from Bieber et al. (2020). These programs include small integers (0, 1, and 2), simple while loops, and if statements. We construct a corpus of synthetic programs to mimic the size of the MBPP dataset in Austin et al. (2021), with 400 training programs, 100 validation programs, and 200 test programs. For each program, three random integer inputs are sampled from the range 0 to 9.

We test execution and scratchpad tracing under both few-shot and fine-tuning conditions. For few-shot experiments, the prompt contains three examples of previous tracing problems, as shown in Appendix C. For fine-tuned experiments, we fine-tune models to convergence on the training split, as judged by validation perplexity.

For the few-shot scratchpad experiment, we noticed that models would not assign the variable name output to the final value in the trace, and would instead continue using v0 (the name of the variable returned in the function f) as the variable name for the final output line. We therefore modified the accuracy criterion from checking whether the value of output in the last line of the trace is correct, to checking whether the value of v0 is correct. (Under naive scoring, the few-shot tracing accuracy is roughly zero.) An example of this behavior is shown in Appendix D.

**Results** Table 2 shows our results on synthetic Python problems. In both few-shot and fine-tuned settings, the scratchpad tracing technique leads to higher overall execution accuracy on the 200 test problems. Fine-tuning also improves performance more for the scratchpad tracing technique than it does for direct execution.

## 5.2 SCRATCHPAD BEATS DIRECT EXECUTION FOR REAL PROGRAMS

In our second set of experiments, we explore how well the scratchpad performs compared to execution on real data. Our main evaluation dataset is the MBPP dataset, introduced in Austin et al. (2021). MBPP consists of 1000 programming problems, each of which contains a natural language specification, a ground-truth Python program, and three input-output test cases. These programs involve computation using a large variety of types, including ints, strings, floats, dictionaries, tuples, and more, and include many language features and control-flow structures, such as loops, comprehensions, library imports, API calls and recursion. The evaluation split of the MBPP dataset contains 500 tasks. In order to separate out effects of the generation window size, we report all evaluation metrics on the subset of these tasks for which the ground-truth trace fits within the generation window of the model for all three of the input-output examples. This leaves a subset of 212 test

Table 3: Comparison of models fine-tuned on different data sets and evaluated on MBPP programs. We report "per-task" execution and tracing accuracies, which require all examples to be correctly executed/traced. We additionally report "per-example" accuracies, which correspond to the total fraction of test examples which are executed/traced correctly across the dataset. We find that training scratchpad models on an dataset augmented with samples from the model significantly improves performance for the scratchpad model, while it harms the direct execution model. Combining tracing training data from several sources further improves scratchpad model performance.

| | Direct execution | | Scratchpad | | | | |
| | MBPP | MBPP-aug | MBPP | MBPP-aug | MBPP-aug +CodeNet +single line | MBPP-aug +CodeNet | MBPP-aug +single line |
| | (§5.2.1) | (§5.2.2) | (§5.2.1) | (§5.2.2) | (§5.3) | (§5.3) | (§5.3) |
| per-task execution acc: | 10.3 | 5.1 | 5.1 | 17.3 | **26.6** | 25.2 | 23.4 |
| per-task trace acc: | n/a | n/a | 0.9 | 13.1 | **24.6** | 22.0 | 21.5 |
| per-example execution acc: | 22.0 | 12.3 | 24.6 | 35.5 | **46.0** | 45.3 | 43.5 |
| per-example trace acc: | n/a | n/a | 6.7 | 26.8 | 41.9 | **42.1** | 40.2 |

tasks. Increasing generation and context window length is an important issue for Transformer-based models, but we view it as orthogonal and leave it for future work.

### 5.2.1 PERFORMANCE IS POOR IN THE VERY-LOW-DATA REGIME

In our first experiment with the MBPP data, we train a scratchpad tracing model on the 374 training tasks (3 examples per task, so 1122 overall examples). We discard all training examples which exceed the context window. We compare overall execution results against a model trained on the same 374 training tasks to perform direct execution. The columns labeled "MBPP" for Direct Execution and Scratchpad in Table 3 show the results of this experiment. Neither the scratchpad model or the direct execution model achieve good performance (5% and 10% output accuracy, respectively), and direct execution outperforms the scratchpad model.

### 5.2.2 SAMPLED PROGRAMS MAKE GOOD SCRATCHPAD TRAINING DATA

Next, we employ a data augmentation technique to increase the size of the training dataset: We first run few-shot synthesis on the 374 MBPP training tasks using the pre-trained 100B+ model, as described in Austin et al. (2021). For each task, we sample and record 80 candidate programs $\{P_s\}$ from the model at temperature $T = 0.5$. We can then create a new execution datapoint using the candidate program $P_s$, the original three inputs for the task $\{x_i\}_{i=1,2,3}$, and the three new outputs which result from running the candidate program on the original three inputs: $\{y_{i_{\text{new}}}\}_{i=1,2,3}$, where $y_{i_{\text{new}}} = P_s(x_i)$. We discard any candidate programs for which execution results in an error. Note that the outputs of $y_{i_{\text{new}}}$ may or may not be equal to the original outputs, depending on the computation performed by the generated program $P_s$. Therefore, this augmented direct execution dataset has both additional new programs and new outputs compared to the original dataset. We can analogously create a tracing dataset for our scratchpad model by tracing the execution of each candidate program $P_s$ on each $x_i$. This process produces much larger tracing and execution datasets with 17k new programs, which we refer to as **MBPP-aug**.

Conceptually, we have augmented the dataset using a combination of tools already available to us, namely a) the neural model, and b) program execution via a Python interpreter. We fine-tune direct execution and scratchpad models on this new augmented dataset MBPP-aug, using the same process as above.

The "MBPP-aug" columns in Table 3 show the results of this experiment. While the direct execution approach suffers a decrease in accuracy when trained on this additional data, the performance of the scratchpad model is greatly improved; the model trained on the augmented data solves more than three times the number of tasks as the model trained on only the original MBPP programs. We also note that if we measure the raw correctness across samples, the model already achieves 26.8% exact trace match, which is surprisingly high.

### 5.3 SCRATCHPAD TRAINING MAKES GOOD USE OF LARGE DATASETS

```
state: b = 15; code: b = b // 2; output: b = 7;

state: g = 100; i = 1; l = [100, 100, 0, 0, -100, -100];
    code: g += l[i]; output: g = 200; i = 1; l = [100, 100, 0, 0, -100, -100];

state: s = 'aabbcd'; code: o = set(s); output: o = {'a', 'b', 'c', 'd'}; s = 'aabbcd';

state: f = 63; i = 11; j = 53; code: f = i ^ j; output: f = 62; i = 11; j = 53;
```

```python
a, b, x = map(int, input().split())
if a // 2 < b:
    if x % 1000 == 0:
        print(a*(x//1000))
    else:
        if (x % 1000) / 500 > 1:
            print(min(a*(x//1000 + 1), a*(x//1000) + b*2))
        else:
            print(min(a*(x//1000 + 1), a*(x//1000) + b))
else:
    if x % 500 == 0:
        print(b*(x//500))
    else:
        print(b*(x//500 + 1))
```

Figure 6: Top: examples of single line data. Bottom: example CodeNet submission.

In this section, we examine whether collecting additional tracing data from human-written programs further improves tracing performance. This will allow us to understand whether the tracing procedure here is likely to scale well when slightly out-of-distribution tracing data is added to the fine-tuning set. We experiment using two datasets:

**Single-line programs**  This dataset consists of roughly 9 million examples of single-line Python transformations. Figure 6 (Top) shows examples of these transformations. Each transformation consists of an initial set of variables and corresponding values, a single line of Python (together these form the input), and the new set of variables and values which results from running the line (the target). When training on single-line data, we do not introduce intermediate scratchpad steps. While this dataset does not provide examples of the high-level, multi-line control flow of a trace, the data provides good supervision for modeling the execution of individual lines of code, which is a key component of tracing. This data was collected by Fraser Greenlee, and can be accessed here.

**CodeNet**  The Project CodeNet dataset (Puri et al., 2021) consists of millions of user submissions to approximately 4,000 coding problems. These submission include both correct and incorrect solutions to programming problems. However, from the experiment with MBPP-aug above, we know that incorrect or broken programs can still provide a useful training signal. We additionally improved our tracing technique to allow tracing programs with errors; when an error is reached, the error message is added to the end of the trace text and tracing is stopped. We extracted a total of 670,904 traces from the CodeNet data. For each dataset, we first fine-tune the model on these datasets, and then perform a second fine-tuning on MBPP-aug until convergence.

**Results**  Results are shown in Table 3. As above, we report execution accuracy across tasks. We additionally report trace accuracy across tasks, to understand the extent to which the entire trace is accurately predicted. We also report the raw execution and trace accuracy across all test examples, as an additional metric to compare models.

Training on either the single-line dataset or the CodeNet dataset alone seem to both provide gains over MBPP-aug (23.4% and 25.2% tasks executed correctly, respectively). However, combining both CodeNet and the single-line dataset seems lead to the highest performance; tracing produces the correct final output for 26.6% of the tasks, and nearly a quarter of the tasks (24.6%) are traced perfectly for all three examples. These results seem promising: the neural network can often exactly trace programs. In particular, greedily decoding from the best model produces the exact correct trace for almost 42% of all traces.

## 6  RELATED WORK

The tasks in this paper can be viewed as exploring one criticism of large language models, namely, to what extent do they simply rely on surface-level statistical correlations on text, without learn-

ing semantics or world knowledge (Bender & Koller, 2020)? In response, Li et al. (2021) provide evidence that pre-trained language models do indeed construct approximate representations of the semantics of the situations they describe in text. In the context of programs, Austin et al. (2021) approach this question by exploring the *learning to execute* task on MBPP, which we consider in Section 5.2. The idea behind this task was to explore whether neural models for synthesis that generate code could also execute it. While that work finds existing models perform poorly at predicting execution, we show that adding a scratchpad allows these models to perform better.

Work in *learning to execute* has considered whether off-the-shelf recurrent neural networks (Zaremba & Sutskever, 2014) or more specialized architectures (Dehghani et al., 2018; Bieber et al., 2020; Wang et al., 2020) have an inductive bias that is sufficiently well suited for executing and reasoning about arbitrary code. The related problem of *neural algorithm induction* has attracted considerable interest (Graves et al., 2014; Kurach et al., 2016; Kaiser & Sutskever, 2016; Graves et al., 2016; Reed & de Freitas, 2016; Veličković et al., 2020a;b). This work proposes new neural architectures, inspired by theoretical models of computation, whose inductive bias allows them to more easily learn algorithm induction tasks. Several methods for algorithm induction specifically add adaptive computation time to sequence models (Graves, 2016; Dehghani et al., 2018; Banino et al., 2021). In particular, universal transformers include adaptive computation time, and are evaluated both on algorithm induction and on learning to execute tasks (Dehghani et al., 2018). In contrast, a scratchpad is a simple way both to provide a transformer model with adaptive computation time, and also to provide supervision about how to use that additional computation, without requiring modification to the underlying architecture.

Algorithm induction has also been connected to pre-trained models. Lu et al. (2021) show that Transformers can be used to some extent as universal computation engines, by pre-training on natural language, and fine-tuning a small fraction of the weights on non-language tasks, including simple algorithm induction tasks. Finally, supervised approaches to semantic parsing (Zelle & Mooney, 1996; Zettlemoyer & Collins, 2005; Kwiatkowksi et al., 2010; Wong & Mooney, 2006) predict the text of a database query, which can then be executed to answer a natural language question.

## 7    LIMITATIONS AND FUTURE WORK

**Context window size**    In this work, we limit our experiments to problems where the scratchpad text fits within the model generation window (512 tokens). However, many problems require very long scratchpad generations. Therefore, fully realizing the potential of the scratchpad technique may require further improvements in transformer generation window size. This is an active area of research in NLP (Tay et al., 2020), and improvements would be beneficial for the scratchpad technique.

**Learning to use the scratchpad without supervision**    A clear next step is to try to learn to use the scratchpad without direct supervision. A simple method would be to use reinforcement learning (RL) techniques: models would be rewarded for correctly answering questions, with reward inversely proportional to the number of scratchpad tokens used. We would hope that learning to use the scratchpad would be a transferable skill; for example, a model could potentially use the algorithm it learned to perform long addition to succeed at polynomial evaluation.

## 8    CONCLUSION

In this work we showed—through experiments on long addition, polynomial evaluation, and Python code execution—that allowing models to read from and write to a simple scratchpad can improve their performance on algorithmic tasks. Such models may be a first step toward combining the knowledge-compression capabilities of large language models with reasoning capabilities, in order to build models that *understand* code as well as write it. This could be useful for a variety of applications that require both working with natural language and reasoning about program semantics, such as program synthesis, neural-guided program analysis, and interactive programming assistants. The scratchpad technique presented here might not take us all the way toward that goal, but we hope it is an important step.

## REPRODUCIBILITY

All fine-tuning and evaluation datasets used in this work are either open source or synthetic and easily reproducible (this excludes MBPP-Aug, which depends on generations from the pre-trained transformer model used in this work). Examples of exact prompts are shown in the Appendix, so that they can be exactly reproduced. Although the pre-training details are not open-source, they correspond to the details in Austin et al. (2021).

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

## A    EFFECTS OF SCRATCHPAD EXECUTION TRAINING ON SYNTHESIS PERFORMANCE

To measure the extent to which fine-tuning on the tracing task described above affects the model's ability to perform program synthesis, we ran a few-shot synthesis experiment using the "MBPP-aug + CodeNet + single line" model. Specifically, we performed few-shot synthesis on the MBPP dataset, as described in Austin et al. (2021). For each MBPP synthesis task, 80 candidate programs are sampled from the model ($T = 0.5$), and the task is considered solved if any of the candidate programs satisfy all three test cases. For more details, see Austin et al. (2021). The "MBPP-aug + CodeNet + single line" model achieved an overall synthesis accuracy of 54%, compared to the 62% accuracy of the original few-shot model in Austin et al. (2021). This indicates that the scratchpad execution training does not completely disrupt the model's ability to perform other few-shot tasks.

## B    LONG ADDITION ABLATION STUDY

In our long addition experiments in Section 3, we compared a model that was trained to perform "direct execution" (the baseline) vs a model trained to use a scratchpad. Since the model trained to use the scratchpad gets an additional signal from all the intermediate steps shown, we also study what happens if the scratchpad model is subsequently trained to perform direct execution (i.e., directly output the target without using the scratchpad). The result is shown in Figure 7 where we followed the same training procedure as for the original direct execution baseline and scratchpad models. We see no significant benefits from doing any intermediate training using a scratchpad. This indicates that the extra training-time information seen by the scratchpad model does not seem solely responsible for the scratchpad model's improved performance.

## C    EXAMPLE FEW-SHOT PROMPT FOR SYNTHETIC PYTHON EXPERIMENTS

Below is an example of a prompt for few-shot synthetic Python synthesis problems:

```
Consider the following Python function:

def f(v0):
  v0 += 0
  v4 = 2
  while v4 > 0:
    v4 -= 1
    v0 *= 2
  return v0

output = f(6)

What is the execution trace?

[BEGIN]

state: {}
line: def f(v0):
state: {"f": "<callable_object f>"}
```

```
line: output = f(6)
state: {"v0": 6}
line:    v0 += 0
state: {"v0": 6}
line:    v4 = 2
state: {"v0": 6, "v4": 2}
line:    while v4 > 0:
state: {"v0": 6, "v4": 2}
line:      v4 -= 1
state: {"v0": 6, "v4": 1}
line:      v0 *= 2
state: {"v0": 12, "v4": 1}
line:    while v4 > 0:
state: {"v0": 12, "v4": 1}
line:      v4 -= 1
state: {"v0": 12, "v4": 0}
line:      v0 *= 2
state: {"v0": 24, "v4": 0}
line:    while v4 > 0:
state: {"v0": 24, "v4": 0}
line:    return v0
state: {"f": "<callable_object f>", "output": 24}

[DONE]

Consider the following Python function:

def f(v0):
  v0 -= 0
  v0 += 2
  v0 -= 0
  return v0

output = f(4)

What is the execution trace?

[BEGIN]

state: {}
line: def f(v0):
state: {"f": "<callable_object f>"}
line: output = f(4)
state: {"v0": 4}
line:    v0 -= 0
state: {"v0": 4}
line:    v0 += 2
state: {"v0": 6}
line:    v0 -= 0
state: {"v0": 6}
line:    return v0
state: {"f": "<callable_object f>", "output": 6}

[DONE]

Consider the following Python function:

def f(v0):
  v0 -= 0
  v8 = 2
  while v8 > 0:
    v8 -= 1
    v0 *= 1
  return v0

output = f(4)

What is the execution trace?

[BEGIN]

state: {}
line: def f(v0):
state: {"f": "<callable_object f>"}
line: output = f(4)
state: {"v0": 4}
line:    v0 -= 0
state: {"v0": 4}
line:    v8 = 2
state: {"v0": 4, "v8": 2}
line:    while v8 > 0:
```

```
state: {"v0": 4, "v8": 2}
line:     v8 -= 1
state: {"v0": 4, "v8": 1}
line:     v0 *= 1
state: {"v0": 4, "v8": 1}
line:   while v8 > 0:
state: {"v0": 4, "v8": 1}
line:     v8 -= 1
state: {"v0": 4, "v8": 0}
line:     v0 *= 1
state: {"v0": 4, "v8": 0}
line:   while v8 > 0:
state: {"v0": 4, "v8": 0}
line:   return v0
state: {"f": "<callable_object f>", "output": 4}

[DONE]

Consider the following Python function:

def f(v0):
  v0 += 2
  v0 *= 2
  v0 *= 1
  return v0

output = f(8)

What is the execution trace?

[BEGIN]
```

## D  INCORRECT OUTPUT FOR SYNTHETIC PYTHON PROBLEMS

Below is an example of the failure of the few-shot synthetic experiments discussed in Section 5.1. Given a problem such as:

```
Consider the following Python function:

def f(v0):
  v0 += 2
  v0 *= 2
  v0 *= 1
  return v0

output = f(8)

What is the execution trace?
```

The model produces the following trace:

```
state: {"v0": 8}
line: def f(v0):
state: {"v0": 8, "f": "<function f at 0x00BFC1C0>"}
line: v0 += 2
state: {"v0": 10}
line: v0 *= 2
state: {"v0": 20}
line: v0 *= 1
state: {"v0": 20}
line: return v0
state: {"f": "<function f at 0x00BFC1C0>", "v0": 20}
```

Note that this trace is exactly correct, except for the last line: where the model predicted "v0": 20, the correct output is "output": 20. Because this type of error consistently occurs in the few-shot synthetic Python experiments, we modified the evaluation script slightly to consider this output to be correct.

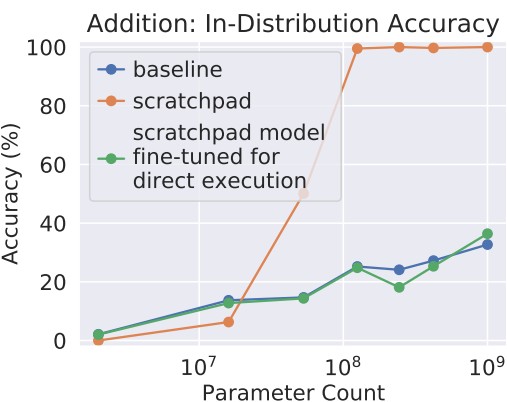

Figure 7: Long addition ablation results. Here, we comparing the baseline and scratchpad results to a model that is first fine-tuned on the scratchpad and then subsequently fine-tuned to perform direct execution (the baseline). The intermediate scratchpad training seem to not have any significant effect on the overall performance, indicating that the extra training-time information seen by the scratchpad model does not seem solely responsible for the scratchpad model's performance.

