# OpenReview forum: "Show Your Work: Scratchpads for Intermediate Computation with Language Models"
_ICLR.cc/2022/Conference — ICLR 2022 Submitted_

### Official Review · Reviewer_WdDB · 2021-10-21

**Correctness:** 1
**Technical Novelty And Significance:** 2
**Empirical Novelty And Significance:** 3
**Recommendation:** 3
**Confidence:** 5

**Main Review:**

## Pros:
* Training models using a scratchpad provides significant empirical improvements in learning to execute

## Cons:
* The explanation of the contribution of the scratchpad seems incorrect to me. I believe that the explanations that the authors present as to why training with scratchpad supervision helps are only partial, and the actual explanation is different (see below). No ablation study was conducted to separate different sources of contribution and explanations.
* The motivation for the task is unclear to me. What use do we have in a far-from-perfect learned python interpreter? Can improvements in this task generalize to other tasks? (see below)
* A direct cost of using a scratchpad is that the output sequence is much longer, making some of the examples in the dataset not fit within the Transformer's context window. At first, I didn't think that this is a practical limitation, but if I understand correctly, this leaves only 212 of the 500 test tasks? This limits the evaluation mostly to short programs and poses scaling limitations on the approach itself.

# Why scratchpads help
The paper provides two possible explanations as to why scratchpads help models perform better, but in my opinion, the paper ignores the main factor.

Starting from the Abstract, the paper lists the challenges in learning to execute programs and arithmetic calculations:

## 1. unbounded computation?
In previous approaches:
>"The model is asked to perform these tasks in one forward pass. Given a fixed number of layers and a fixed amount of computation time, the model cannot adapt the amount of compute spent on a problem to its difficulty before producing an output"

In contrast, scratchpads:
> "Allow the model to produce an arbitrary sequence of intermediate tokens, which we call a scratchpad, before producing the final answer".
> ... "The model has adaptive computation time, that is, it can now process the information for as long as needed"

This gives the impression that the ability to emit tokens that are not included in the final prediction, using a scratchpad, allows unbounded computation.
However, although training with scratchpads *allows* unbounded computation, the models do not perform unbounded computation in practice.

It is theoretically true that the model has an "adaptive computation time" and "can process the information as long as needed", because the model can theoretically do that at test time. But this is also not-true, because the output length is about the same length as the "clean" output. The model is not _trained_ to process the information for as long as needed, and it thus is not likely to do that at test time.  As shown in Figure 1, even "for" loops are not unrolled at training time, but instead, only read once. That is, the total output length is in the order of the original target length, and is not expected to be longer at test time.

## 2. The ability to write to and read from the scratchpad?

The authors hint that the **main** contributing factor to the improved empirical results of using a scratchpad is allowing to emit intermediate computations and reading from them later.

The Conclusion section concludes that:
> "allowing models to read from and write to a simple scratchpad can improve their performance on algorithmic tasks".

This may be one contributing factor and an interesting idea, but the paper did not convince me that it is the main contributing factor.
To empirically prove that this is the most important factor, I would expect experiments that ablate (a) the ability to write to a scratchpad, (b) the ability to perform longer computations, and (c) the additional supervision that is provided at training time.

## The most important factor
While each of (1) (unbounded computation) and (2) (the ability to write to the scratchpad) is a potentially useful ability, I believe that none of them is the main factor that contributes to the improved accuracy of the models that were trained with a scratchpad.
The paper ignores a third factor: the additional supervision signal. The models with scratchpads are trained with the full program execution trace, while the baselines are not. Is it a fair comparison? I am not sure that comparing models that were trained using only the "static" source code and models that were trained with additional "dynamic" execution trace supervision is a fair comparison.

The explanation and presentation of why scratchpads help seem incorrect to me. The authors claim that it's the unbounded computation and the ability to write to a scratchpad, while ignoring the additional supervision. No ablation experiments in the paper have attempted to tease apart these factors and understand the most important sources of contribution.

Is the dependence on this additional supervision information realistic? If this additional supervision source (the step-by-step execution trace) was a new source of supervision that could be utilized in other models of programming languages, then maybe it was useful to use it at training time (if the paper explained it as such!). However, I don't see how this is a source of supervision that can be generalized to other tasks, because it is specific to this task of learning to execute short, standalone, functions.
In most scenarios and tasks, a model cannot execute long programs, partial programs, incomplete programs, programs without test cases, or programs that require complex objects as inputs.

# Unclear task motivation
The motivation of why to learn to execute python programs and arithmetic addition is unclear to me. While it may be intellectually pleasing to experiment with tasks that are too difficult for neural networks to perform and try to see what components are needed to make the neural networks be more and more accurate, I don't see how this transfers to anything practical.

At best, the model just learns to mimic the python interpreter (or the "addition" algorithm).
What benefit do we have in having a neural network that simulates (a very specific kind of short functions on) the python interpreter, which we already have and can execute programs perfectly? What use do we have in a model that has learned to partially simulate the python
interpreter? if we have a python interpreter to provide the execution trace, why do we need a model to learn to execute?

As the authors write in Section 5.2.2:
> "we have augmented the dataset using a combination of tools already available to us, namely ... program execution via a Python interpreter".

This is the point - if we consider the python interpreter as a tool that is already available to us, why do we need to learn it? Is this a proxy task to other tasks? Can improvements in this task generalize to other tasks?

## Additional Questions:
1. As shown in Figure 3, the difference between the scratchpad model and the baseline model starts to affect the performance only starting from sufficiently large models. In smaller models (<50M?), there seems to be no difference, even for the task of arithmetic addition. This is not discussed in the paper. Can the authors explain this phenomenon? I would expect that even a small model (and even other architectures such as LSTMs) would benefit a lot from the additional scratchpad supervision.

2. "direct execution prediction requires the model to correctly output the result of executing the entire function in a *single pass*" (section 5, page 5). But then: "we train models to predict an alternating sequence of 1) the ordered sequence of source code lines executed, and 2) the state of the local variables after each line is executed". Isn't this also a *single pass*?

3. I do not understand the message of the "single-line programs" discussed in Section 5. As shown in Table 3, further training on this additional dataset improves the results even more. But what is the message here? that training on more data improves accuracy?

4. "neural networks that **understand** code" is an incorrect, confusing, and harmful terminology:

>"So language models can to some extend _write_ code, but do not seem to **_understand_** the code they write ..."

>"Neural networks that **understand** programs could enable..."

> "Such models may be a first step toward ... in order to build models that **_understand_** code"

This terminology contributes to the hype over neural networks but confuses the audience that might be given the wrong impression and an imprecise understanding of neural networks and how they work.

5. The paper states that:
>"Why do large language models struggle with algorithmic reasoning tasks? We suggest that this is at least partly due to a limitation of the Transformer architecture"

But later, the paper does "exploit existing Transformer architectures" ... "without modifying the underlying architecture". So, do the authors argue that the Transformer architecture inherently limited or not?

# Missing related work:
The recognition that providing step-by-step supervision improves execution prediction was demonstrated in some previous work as well. The following recent papers were not discussed nor compared to:
* Veličković et al, "Neural Execution of Graph Algorithms", ICLR 2020
* Veličković et al, "Pointer Graph Networks", NeurIPS 2020
* Yan et al., "Neural Execution Engines: Learning to Execute Subroutines", NeurIPS 2020
* Wang et al., "Dynamic Neural Program Embedding for Program Repair", ICLR 2018
* Wang et al., "Blended, Precise Semantic Program Embeddings", PLDI 2020
* Wang et al., "Learning Semantic Program Embeddings with Graph Interval Neural Network”, OOPSLA 2020

I agree that these papers did not use a scratchpad, but I was not convinced that the scratchpad itself is the most contributing factor in this work, and no ablation study showed that.
I also agree that some of these address learning graph algorithms (and not python programs), but I would also expect a conceptual comparison and discussion.



**Summary Of The Paper:**

This paper addresses the "learning to execute" task, of predicting the output of a given python program or arithmetic calculation. The main novelty is to train Transformers with "scratchpads": At training time, the model is provided with the intermediate states of all variables after executing every statement; at test time, the model predicts these intermediate states but is allowed to mark them as part of its "scratchpad". These intermediate states are not included in the final prediction, but only serve as a temporary output buffer.


**Summary Of The Review:**

Although training a model using a scratchpad provides significant empirical improvements in learning to execute, and I appreciate the extensive empirical evaluation conducted by the authors, I believe that the explanation of why scratchpad helps is incorrect and confusing the readers. Further, the motivation of this task and settings of learning the python interpreter given a python interpreter at training time, is unclear to me.

Thus, I recommend rejection at this time, but I am open to changing my mind if convinced that the main weaknesses listed above are mistaken.

---

> ### Author Response · Authors · 2021-11-18
> **Response to Reviewer WdDB Part 1**
>
> Thank you for the comprehensive review---our responses to individual comments/questions are below.
>
> **As shown in Figure 1, even "for" loops are not unrolled at training time, but instead, only read once.**
> We would like to point out a potential confusion: at training time, the model is actually trained to fully unroll loops. We apologize for this confusion, which is due to the fact that both of the loops in Figure 1 each only happen to iterate once. We will update the figure to illustrate this better and make it less confusing.
>
> **That is, the total output length is in the order of the original target length, and is not expected to be longer at test time.**
> Yes, we agree that we could have been crisper about what we meant here. What we mean is that the transformer can learn to do “one step” of computation in the forward pass that then gets unrolled/repeated across time. We will update for clarity.
>
>
> **The paper ignores a third factor: the additional supervision signal. The models with scratchpads are trained with the full program execution trace, while the baselines are not.**
> We ran an ablation experiment that we think should (at least partly) satisfy your concern about this:
> On the long-addition task, we first fine-tuned the model to use the scratch pad, as in the original experiment. Then we took that same fine-tuned model (notice that this model has now seen all of the same training signal as the scratchpad model) and re-fine-tuned it to directly output the answer. The result for the “direct execution” experiment is identical to the one in the paper, which suggests that merely having seen all the additional bits contained in the scratch-pad computations is not by itself useful. We are adding this experiment to the paper.
>
> We think this is reasonably convincing, and that it makes sense. Our intuition says that the thing that really matters is the number of pairs ((x,y), z = x + y), not the number of tokens seen.
>
> As another argument against this point, consider that the few-shot results for both polynomial evaluation and synthetic program execution show that the scratchpad helps a lot. Since these results are few-shot, there is no real way that extra information about the tasks could have been compressed into the model weights.
>
> **This is the point - if we consider the python interpreter as a tool that is already available to us, why do we need to learn it? Is this a proxy task to other tasks? Can improvements in this task generalize to other tasks?**
> We could have been clearer about this. Let us point to what another reviewer said:
> >1. The main scratchpad idea presented here is new and interesting, not only to people who care about executing programs with LMs but also to the language modeling community at large.
> >If we can figure out how to make language models better understand python, that might lead us in a direction that will enable us to build LMs that can better understand natural language.
> >Intermediate computations are not only a property of programming languages, but have lots of relevance to natural language as well, see for example the wide range of research on multi-hop question answering.
> 2. I’m very happy to see language modeling research that improves results without actually changing the language model itself! I think this opens up a lot of possibilities that were not available before.
> >With LMs growing in size, it is becoming less practical for researchers to train them from scratch for each project.
> >I hope that this paper motivates the community to further explore research directions which don’t require model modifications at all.
>
> We would add to this that we think the current work is an important intermediate step toward getting these models to do multi-step “reasoning” more generally, which is our broader goal.
>
> **Can the authors explain this phenomenon?**
> We don’t think we have a particularly satisfying explanation here. As others have noted, sometimes there is a specific place on the size-curve after which models quickly acquire new behaviors. We speculate that the critical size to do the pattern matching required for scratchpad use is around 100 million parameters, but again we don’t have a very refined theory for why. We believe investigating these types of phenomena are very promising directions for future work.
>
> **Isn't this also a single pass?**
> We meant that in the direct execution case --- because you need to emit the output of the program as soon as you've seen the input --- you have to execute the whole program in one pass. If you use the scratchpad, then because you are sampling one line at a time, you get to use the neural network to execute a single line, rather than a whole program. Certainly not all lines represent the same level of abstraction, and figuring out how the model can dynamically decide to “drill down on” a line is important future work! We will make this more clear.

---

> > ### Author Response · Authors · 2021-11-18
> > **Response to Reviewer WdDB Part 2**
> >
> > **I do not understand the message of the "single-line programs" discussed in Section 5.**
> > Executing a program involves figuring out which line to go to next and figuring out what each line does. The nice thing this experiment shows is that you can train on examples of only the latter and get better results on the overall execution problem. That is, there is some compositionality going on here.
> >
> > **"neural networks that understand code" is an incorrect, confusing, and harmful terminology**
> > We agree that this could be confusing, and will change it.
> >
> >
> > **But later, the paper does "exploit existing Transformer architectures" ... "without modifying the underlying architecture". So, do the authors argue that the Transformer architecture inherently limited or not?**
> > We do argue that it is inherently limited, and that we have made it perform better at a certain task by, in some sense, changing the task. A more precise wording of this is that the architecture and task definition together are what is flawed. We will make this change.
> >
> > **I also agree that some of these address learning graph algorithms (and not python programs), but I would also expect a conceptual comparison and discussion.**
> > We will add a discussion of these papers, thank you for bringing them to our attention!

---

> > > ### Comment · Reviewer_WdDB · 2021-11-29
> > > **Response to authors**
> > >
> > > Thank you for your response.
> > >
> > > ### Why scratchpads help
> > >
> > > > Our intuition says that the thing that really matters is the number of pairs ((x,y), z = x + y), not the number of tokens seen
> > >
> > > I am not sure I agree. The number of tokens seen correlates with the number of cross-entropy loss terms, so why can it be canceled out so easily?
> > >
> > > Also, I think that what really matters is the kind of information contained in these tokens, and not only their number.
> > >
> > > ### No downstream task
> > >
> > > I was still not convinced that this training with a scratchpad is useful without a real downstream task. We all agree that learning the python interpreter itself is not useful. The authors argue that learning this would be useful for other, future tasks. However, the paper does not show that. When I asked, the authors quoted other reviewers instead of answering themselves:
> > >
> > > > We could have been clearer about this. Let us point to what another reviewer said ...
> > >
> > > I expected a more in-depth explanation than just quoting other reviewers.
> > >
> > > The authors have only argued that this is "an important intermediate step toward getting these models to do multi-step reasoning".
> > > The paper could have been insightful if it really showed that learning the interpreter is useful for other downstream tasks. As long as the paper does not show that, we have no way to tell whether this is an important step or not.
> > >
> > > ### Size of context window limitation
> > >
> > > In my review, I wrote that:
> > >
> > > > A direct cost of using a scratchpad is that the output sequence is much longer, making some of the examples in the dataset not fit within the Transformer's context window... if I understand correctly, this leaves only 212 of the 500 test tasks?
> > >
> > > I am aware that this was not phrased as an actionable question, but the authors completely ignored this comment+question. Reviewer uxzZ also wrote that:
> > >
> > > > the MBPP dataset the authors had to prune more than 50% of the tasks since they were too long to fit entirely within one inference pass of the transformer model used. So the results are biased towards shorter programs
> > >
> > > and the authors ignored this comment as well.
> > > Actually, not only that the results are biased towards programs that are shorter in their code, the results are biased towards programs that are shorter in their runtime. It looks like scratchpads are limited only for programs with `O(n)` complexity at most.
> > >
> > > I think that a thorough discussion and analysis of how much training with a scratchpad increases the length of the examples is crucial for understanding whether this could be really practical.
> > >
> > > Overall, I keep my score. I honestly hope that the authors could show in a future version how training with a scratchpad is useful for _any_ downstream task other than only training the interpreter.

---

> > > > ### Author Response · Authors · 2021-11-29
> > > > **RE: Response to authors**
> > > >
> > > > Thank you very much for your additional comments. Our responses are below.
> > > >
> > > > **Why scratchpads help**
> > > > In our initial response, we performed an experiment in the long addition domain to help address your concerns about why scratchpads lead to improved performance (see above response, and Appendix B of the revised draft). Please let us know if you have any questions about this experiment, and if there are reasons that it does not address your concerns.
> > > >
> > > > **I expected a more in-depth explanation than just quoting other reviewers.**
> > > > We thought it was stronger to respond in the words of the other reviewers, not our own words, to show that experts other than the authors of the paper agree that this work has value to the broader community.
> > > >
> > > > **Size of context window limitation**
> > > > Yes, the results are biased towards shorter programs, because context/generation windows have limited size (which is true for any experiment with transformer models). However, the positive affect of the scratchpad approach still persists. We also believe that the context-window question is a practical limitation of current models, orthogonal to our research questions, and is an active area of research (e.g., Press et al., 2021; Beltagy et al., 2020).
> > > >
> > > > Press et al., 2021: https://arxiv.org/abs/2108.12409
> > > > Beltagy et al., 2020: https://arxiv.org/abs/2004.05150

---

### Official Review · Reviewer_uxzZ · 2021-10-27

**Correctness:** 3
**Technical Novelty And Significance:** 4
**Empirical Novelty And Significance:** 4
**Recommendation:** 8
**Confidence:** 4

**Main Review:**


Strengths:

1. The main scratchpad idea presented here is new and interesting, not only to people who care about executing programs with LMs but also to the language modeling community at large. If we can figure out how to make language models better understand python, that might lead us in a direction that will enable us to build LMs that can better understand natural language. Intermediate computations are not only a property of programming languages, but have lots of relevance to natural language as well, see for example the wide range of research on multi-hop question answering.
2. I’m very happy to see language modeling research that improves results without actually changing the language model itself! I think this opens up a lot of possibilities that were not available before. With LMs growing in size, it is becoming less practical for researchers to train them from scratch for each project. I hope that this paper motivates the community to further explore research directions which don’t require model modifications at all.
3. The ideas presented in this paper are simple and easy to reproduce.
4. The paper is written in a very straightforward and comprehensible way.



Weaknesses:
1. Table 3: I’m guessing that you didn’t train the direct execution model on the following datasets:  MBPP-aug +single line, and MBPP-aug +CodeNet +single line, because you saw that MBPP-aug didn’t help? But why are there no results for the direct execution model trained on MBPP + CodeNet or MBPP + single line? If these datasets help the scratchpad model so much, it would be very relevant and interesting to see how much they help the baseline.
2. This isn’t a major issue but it seems like in the MBPP dataset the authors had to prune more than 50% of the tasks since they were too long to fit entirely within one inference pass of the transformer model used. So the results are biased towards shorter programs. This is totally fine but I felt like the authors should explain that a bit more in the paper and specify exactly how long the context window of their model is so the reader understands how long these programs are.


Other notes/questions:
1. “we aim to reduce the propagation and compounding of small errors, because states are quantized to token embeddings”
It's not clear to me that quantizing would help reduce the compounding of errors. What happens if you output a wrong token? Isn’t that even worse than having a small “error” in some continuous representation space?
2. Table 2 has an asterisk at the bottom left which is not explained in its caption. It would be helpful if you would add that note about the accuracy criterion to that table’s caption as well (if that’s what you mean by that asterisk).
3. There aren’t many details given about the model used. For example, what position embedding method did you use? I wonder if using a relative position method would have improved the model’s OOD abilities on the addition task.


**Summary Of The Paper:**

Transformers LMs are getting better and better at generating programs given a description, but do they actually “understand” what they’re generating? Can transformers execute code? The current state of the art in this domain is weak, and the authors show an incredibly simple method here that leads to big performance gains. Instead of modifying the model or the training data, the authors show that by simply training the transformer to output the result of intermediate computation steps, its performance when executing code significantly goes up.
The authors show strong results on addition and a few other artificial tasks, before moving on to a codebase of python programs. Their scratchpad method improves over the baseline and benefits from a simple data augmentation method that they introduce.


**Summary Of The Review:**

This paper presents a new, simple idea, is well written, and has strong empirical results. The research methodology here, of improving LM performance without actually changing the architecture, is super interesting, and I hope that this paper leads to more work that pushes the state of the art forward without modifying the model. The paper presents results on artificial tasks and python code but I believe that conclusions from it are relevant to the entire natural language modeling field. I am strongly in favor of accepting this work!

---

> ### Author Response · Authors · 2021-11-18
> **Response to Reviewer uxzZ**
>
> Thank you very much for the encouraging review!
>
> Thank you for the comments about the context window, the Table 2 asterisk, and the model details: we will make each of these clearer in the paper.

---

### Official Review · Reviewer_PHbv · 2021-11-04

**Correctness:** 4
**Technical Novelty And Significance:** 3
**Empirical Novelty And Significance:** Not applicable
**Recommendation:** 8
**Confidence:** 4

**Main Review:**

Strengths
- The idea is straightforward and leads to clear improvement in the empirical results.
- The work serves as a nice complement to others which outsource the algorithmic reasoning to outside the machine learning model (e.g. execution-guided program synthesis), and may catalyze further research on the right balance between the work done by the ML model, and by program interpreters or other discrete reasoning engines.

Weaknesses
- For an entirely empirical paper, the tasks considered are not directly useful to solve with a learned model, since there already exist simple and straightforward ways to perform addition, polynomial evaluation, and Python evaluation with computers. It would be nice to see improvements on a more "useful" task.
- There could be more systematic analyses of the settings where the idea works well and where it does not, beyond the ones already in the paper. For example, we can vary the size of the underlying model, the amount of training data, complexity of the task, etc.

Questions
- For the few-shot setting, it was unclear exactly how the few-shot examples were chosen. Were they chosen separately for each test example (e.g. to include the training examples most similar to the test example)? If so, what was the method? If not, what was the methodology used to select the fixed set of examples? How robust are the results to changing the set used?
- The CodeNet paragraph says: "We additionally improved our tracing technique to allow tracing programs with errors; when an error is reached, the error message is added to the end of the trace text and tracing is stopped". Which columns in Table 3 was this improvement applied to? Do we know how much this affects the results, independent of the extra data from CodeNet?


**Summary Of The Paper:**

The paper evaluates the ability of large pre-trained language models to perform addition, polynomial evaluation, and execution of Python programs, with few-shot in-context learning or with fine-tuning. Instead of having the model directly predict the output of the task, the authors propose to have the model generate all of the intermediate steps involved: for addition, the results of adding every column and the carry; for polynomial evaluation, the value for each term; for Python evaluation, the value of all variables after each line. The authors demonstrate improvements compared to direct execution for all of these settings.

The authors propose several reasons that would enable such an improvement:
- Adaptive computation time, as the model runs longer for more complicated tasks
- Intermediate state can be stored in the past tokens rather than only in the activations
- Reduced propagation of errors as states are quantized to specific tokens
- Debugging and interpretability


**Summary Of The Review:**

I think the paper presents an elegant idea with extensive experiments and would provide useful insights to the community.

---

> ### Author Response · Authors · 2021-11-18
> **Response to Reviewer PHbv**
>
> Thank you very much for your positive review! Below are responses to your comments and questions:
>
> - We note that in our long-addition experiments, we showed how the technique performs as model size and task difficulty (in-distribution vs out-of-distribution test data) are varied---see Figure 3.
>
> - For the few-shot setting, the prompt examples were chosen randomly and fixed for all test problems.
>
> - Only the tracing of the CodeNet data was done to allow tracing programs with errors, because a large fraction of the CodeNet traces included errors. We have not measured how much this change independently affects results.

---

### Official Review · Reviewer_UqLx · 2021-11-04

**Correctness:** 2
**Technical Novelty And Significance:** 2
**Empirical Novelty And Significance:** 2
**Recommendation:** 3
**Confidence:** 4

**Main Review:**

*Pros:*

- Overall the paper is easy to follow and the idea of augmenting the output with scratchpads is simple and interesting and is shown to perform better than outputting the values of direct execution of the input.

- The scratchpad method shows promises to benefit better from synthetic data augmentation, than direct execution.


*Concerns & Areas of enhancement:*

- *Limitations of the Fine-tuning scenario:* Majority of the results shown in the paper are on the finetuning setup not the few-shot ICL one. The power of large language models is that they are general in context learners.  Finetuning a large LM of that size will completely destroy this capacity. After all, there's no need of having a 100B parameter model that just adds numbers, solves polynomial equations, or Execute python code of fixed length from an application-wise. If authors see that this finetuning is not as destructive to the model, extra evaluation on language modeling or general zero-shot/few-shot in context learning should have been reported.


- *Comparison against Meaningful baselines* If we accept that the focus of the study here is to learn neural networks that can ONLY perform complex multi-step computations and "scratchpads" are a good method for this. In this case, the authors should have compared meaningful baselines for compositional generalization [1,2,3,4] and aligned with those lines of work.

  * 1- https://arxiv.org/pdf/1711.00350.pdf
  * 2- https://arxiv.org/pdf/2109.12243v1.pdf
  * 3- https://proceedings.neurips.cc//paper/2020/file/83adc9225e4deb67d7ce42d58fe5157c-Paper.pdf
  * 4- https://arxiv.org/pdf/2003.05562.pdf


- *Reproducibility & Transparency in reporting methods and experiments*

The paper lacks lots of reproducibility details on the used models (their pretraining method), finetuning details, few shot examples selections. There's no way to verify the correctness of the result in the paper or to reproduce any of the results if the paper is accepted.
To complete this authors don't provide any reproducibility statement. I advise the authors to read the reproducibility section in the ICLR2022 authors guide https://iclr.cc/Conferences/2022/AuthorGuide



**Summary Of The Paper:**

This paper proposes an output augmentation method for enhancing the capacity of Transformers language models in performing multi-step symbolic computations such as adding integers or executing programs.
The idea in brief is instead of finetuning/ICL the model to directly output the result of the execution of the program, the expected output becomes a "scratchpad" of the execution steps terminated by the execution prediction.
The intuition here is enforcing the output signal has the following advantages 1) allowing adaptive computations time with respect to the input (i.e. the length of the generated scratchpad increases with more complicated inputs) 2) Acting as an intermediate memory 3) Allowing for interpretability.

Authors evaluate the proposed through finetuning or In context Learning of Decoder only transformers (2M->100B params), on three tasks:
- Long integer Addition (finetuning only)
- Polynomial evaluation (Finetuning and ICL)
- Python program execution both on synthetic and real programs (Finetuning only)

Overall, the results show that:
- Transformers trained to output scratchpad in addition to the final prediction, perform better than direct prediction only.
- They could benefit better from synthetic data generation and data augmentation.


**Summary Of The Review:**

Overall the scratchpad idea sounds simple and interesting, however, the current evaluation in the fine-tuning setup has it's practical limitations (see above). I advise the authors to either position this paper in one of the two directions discussed above. Overall the reproducibility and transparency in experiment reporting would benefit from major enhancements.

---

> ### Author Response · Authors · 2021-11-18
> **Response to Reviewer UqLx**
>
> Thank you for the review! Our responses are below.
>
> **Limitations of the Fine-tuning scenario**
> We would like to point the reviewer to Tables 1 and 2 for few-shot results. Using the few-shot scratch-pad more than doubles the performance for both polynomial evaluation and execution of synthetic programs, showing the merit of our approach beyond the fine-tuning scenario.
>
> The reason we mostly study fine-tuned models is that they perform better, and the criticisms of “fine-tuning” methods in this review could be applied to any work which fine-tunes large language models. Besides, we will argue that the claim “Finetuning a large LM of that size will completely destroy [its few shot learning] capacity”  is empirically false in this context: We have results (though not in the original draft) which show that the models fine-tuned on these scratchpad tracing tasks for Python can still perform few-shot synthesis with good accuracy. We will update the draft to report these numbers in the appendix.
>
> **Comparison against Meaningful baselines**
> Thank you for pointing out these additional benchmarks. The current paper focuses on algorithmic capabilities and not on compositional tasks, so we feel that the mentioned baselines are reasonably far out of scope, but they are interesting future work.
>
> **Reproducibility & Transparency in reporting methods and experiments**
> This is a totally fair point, and we will fix it before the end of the discussion period.

---

> > ### Comment · Reviewer_UqLx · 2021-11-30
> > **Reply**
> >
> > Thanks for your replies.
> >
> > Considering reproducibility, I was expecting the authors to more significant efforts on this aspect, instead, authors add just a minimal reproducibility statement that doesn't serve much beyond a few more details.
> >
> > > Although the pre-training details are not open-source, they correspond to the details in Austin et al (2021)
> >
> > Adding the previous statement is not enough for reproducibility. I find the state of transparency of the paper overall is still not adequate for publishing. There's still no way to verify the correctness of the results or the additional results on MBPP. Given this, I still hold my stance with respect to reproducibility
> >
> > > Thank you for pointing out these additional benchmarks. The current paper focuses on algorithmic capabilities and not on compositional tasks, so we feel that the mentioned baselines are reasonably far out of scope, but they are interesting future work.
> >
> > My point here is generally about comparison with meaningful baselines, not just general transformers. "Algorithmic capabilities" entails "compositionality", authors should have put the effort to find meaningful baselines for making a meaningful comparison.
> >
> > > Using the few-shot scratch-pad more than doubles the performance for both polynomial evaluation and execution of synthetic programs, showing the merit of our approach beyond the fine-tuning scenario.
> >
> > While this is true, do you have any reason for not including ICL results for the real python program execution? While there are gains in ICL on those two tasks, these tasks are synthetic.  And due to the lack of transparency in reporting the created datasets (as in the paragraph below), it is impossible to judge the difficulty of such tasks (e.g. in terms of generalization).
> >
> > > " We construct a corpus of synthetic programs to mimic the size of the MBPP dataset in Austin et al. (2021), with 400 training
> > programs, 100 validation programs, and 200 test programs. For each program, three random integer
> > inputs are sampled from the range 0 to 9."
> >
> > **comments on ICL performance**
> > In the additional experiments on MBPP you mention:
> > > "For each MBPP synthesis task, 80 candidate programs are sampled from the model (T = 0.5), and the task is considered solved if any of the candidate programs satisfy all three test cases."
> >
> > The drop in ICL for synthesis (the additional experiment reported in the paper) from 62% to 54%  is quite a significant one this is given that condition above very relaxed  Acc@80. Authors use the term Accuracy to denote Accuracy@80 which is misleading to convince the reader that numbers are higher than they are.
> >
> > Given the reasons above, my evaluation of the paper stays unchanged. I advise the authors to do efforts in terms of transparency wrt reporting results, training details, and overall reproducibility in the paper. Additionally, to make meaningful comparisons and align their work with established tasks in the community rather than positioning it as an isolated island.

---

> > > ### Author Response · Authors · 2021-11-30
> > > **RE: Reply**
> > >
> > > Thank you very much for your response.
> > >
> > > - With respect to the pre-training details, we note that, as stated in the revised text, many details are given in Austin et al., 2021. We reproduce this text below, and hope it helps address questions about transparency:
> > > > The models we use in this paper are dense left-to-right decoder-only Transformer language models [Vaswani et al., 2017] trained on a combination of web documents, dialog data, and Wikipedia. Our experiments were conducted using models with non-embedding-parameter-counts ranging from 244 million to 137 billion. The pre-training dataset for the model contains 2.97B documents, which were tokenized into 2.81T BPE tokens with a vocabulary of 32K tokens using the SentencePiece library [Kudo and Richardson, 2018]. This data included web sites with both computer code and text, such as question and answer sites and tutorials, but source code files themselves were not specifically included, except where code appeared in other web sites. These web sites with code and text comprised about 13.8M documents containing 18.7B BPE tokens out of the pre-training data.
> > >
> > >   Although we agree that it is a benefit to the work and the community when models and datasets themselves can be made public, it is a complex issue when the results from evaluating models that are not publicly available are nevertheless important enough to merit publication.
> > >
> > > - We did not run few-shot experiments on the real python data because, as shown in Table 3, fine-tuned direct execution and scratchpad tracing only achieved 10% and 5% per-task accuracy, respectively. Presumably, few-shot accuracy would be lower than this, which hopefully can help provide a sense for the difficulty of these tasks.
> > >
> > > - We apologize that the way we presented the new results for few-shot synthesis results seems misleading. We note that this is the same terminology used in Austin et al. (2021).

---

### Comment · Reviewer_WdDB · 2021-11-18
**Discussion with authors?**

Dear authors,

The public discussion period is about to end and the authors did not respond to any of the reviews.

Although my initial score is low, I did hope to discuss the paper with you during this discussion period and give the authors the opportunity to convince me that I was wrong in my initial assessment.

---

> ### Author Response · Authors · 2021-11-18
> **RE: discussion**
>
> Thank you very much for this comment and your thorough review.
> We have added individual responses to each review below, and look forward to discussing with the reviewers!

---

### Author Response · Authors · 2021-11-22
**Summary of changes**

We would like to thank all of the reviewers for their encouraging and helpful comments. We have updated the submission, taking into account comments from the reviewers.

Our changes include:
- adding a reproducibility statement (Reviewer UqLx)
- adding a short section in the appendix describing the effect of Python scratchpad fine-tuning on few-shot synthesis performance (Reviewer UqLx)
- adding a short section in the appendix on an ablation study in the long addition domain (Reviewer WdDB)
- adding additional relevant citations (Reviewer WdDB)
- making several additions for clarity, as requested by reviewers (Reviewers UqLx, PHbv, uxzZ, and WdDB)

---

### Decision · Program_Chairs · 2022-01-20

**Decision:**

Reject

**Comment:**

The main remaining criticism of the paper is reproducibility, i.e., "it is nearly impossible to verify the correctness of the result in the paper or to reproduce any of these results" (AC). We generally agree with this statement. While the authors do provide some details in the paper, reviewers, AC, SAC, and PCs agree that this is insufficient. Further points that came up in our discussions were the simplicity of the baselines and the choice of testing to demonstrate that the approach really works. Our impression is that the work lacks a rigorous experimental evaluation. These considerations led to the decision in the end.